# Mating Systems of Single Families and Population Genetic Diversity of Endangered *Ormosia hosiei* in South China

**DOI:** 10.3390/genes13112117

**Published:** 2022-11-15

**Authors:** Fengqing Li, Huanwei Chen, Suzhen Liu, Huacong Zhang, Zhichun Zhou

**Affiliations:** 1Zhejiang Provincial Key Laboratory of Tree Breeding, Research Institute of Subtropical Forestry, Chinese Academy of Forestry, Fuyang 311400, China; 2Experimental Center of Subtropical Forestry, Chinese Academy of Forestry, Fenyi 336600, China; 3Longquan Research Institute of Forestry, Longquan 323800, China

**Keywords:** SSRs, mating system, genetic diversity, biparental inbreeding, *Ormosia hosiei*

## Abstract

*Ormosia hosiei* is a tree species native to China that has been extensively used for ornamental and furniture purposes due to its valued timber. The mating system has substantial impact on genetic diversity and structure of plant natural population. Such information should be considered when planning tree planting for forest restoration. Here we used 12 microsatellite markers and described the mating system of single families and the population genetic diversity of *O. hosiei*. A high level of genetic diversity was observed in both adults and progenies, although slight differences existed among populations and their progenies, with the expected heterozygosity ranging from 0.763 to 0.794. Overall, *O. hosiei* displayed a predominantly outcrossed mating. The estimate of multi-locus outcrossing rate (tm) was high with low variations among families, ranged from 0.997 to 1.000. The value of tm-ts, ranged from 0.000 to 0.139, indicated that biparental inbreeding occurred in progenies. Therefore, to obtain a reasonable genetic representation of native tree species and prevent problems associated with inbreeding depression, we suggested effective in situ conservation by replanting seedlings, but seedling production for restoration purposes may require a much larger sampling effort than is currently used. Moreover, it is necessary to conduct further multiple population and multi-year experiments to verify our conclusions.

## 1. Introduction

Genetic diversity and its genetic structure of a tree population are closely related to life history traits [1], in which the mating system is a critical biological factor, has a significant bearing on the subsequent generations, gene flow and natural selection [2,3,4]. The mating system refers to the mating patterns among fruits within individuals, individuals within populations, among populations and from one flowering event to another, including outcrossed, auto-fertilization and apomictic. To a certain extent, it determines how genetic information is passed between generations and among populations, as well as the offspring fitness [4]. Variation of tree mating patterns at different levels may be influenced by several ecological variables, such as plant density or flowering plant density [5], population size of adult trees [6,7], floral morphology and pollinator activity [8]. Therefore, to the populations in anthropogenic disturbance and requiring restoration or conservation, the understanding of the mating system and factors affecting it is of fundamental importance to genetic structure [9]. Such information is useful in precondition for genetic improvement, population conservation and rational exploitation and utilization of resources.

In the process of mating system evolution, some studies believed that it was influenced by various factors, resulting in the adaptation mechanism to the environment-mixed mating system, while other scholars considered that no stable mixed mating system existed, and the mating system observed in practices was only a transitional state and self-fertilization was an evolutionary dead end [10]. Mating system analyses have indicated possibilities of detecting variations in mating patterns on a micro-scale level [6,11]. Studies revealed that variation occurred at hierarchical levels (among fruits within individuals, individuals within populations, among populations and from one flowering event to other) [11]. Leguminous plants became suitable materials for studying the interaction among self-incompatibility, outcrossing rate and inbreeding decline due to their mixed mating mechanism [12]. Many studies revealed that shrubs or trees natural population of Leguminous were mainly outcrossing, for example, *Robinia pseudoacacia* [13], *Platypodium elegans* [14], *Erythrophleum fordii* [15] and *Enterolobium cyclocarpum* [16].

Traditional methods used for the analysis of mating systems mostly focused on the analysis of morphology and observation of pollinator behavior, which had many limitations owing to the restrictions of trees, such as long time required for offspring to acquire scored markers and inconsistency between phenotypic markers or pollinator behavior and outcrossing rate. With the rapid development of molecular biology, molecular markers, such as isozyme markers SSR, AFLP and RAPD, have become widely used in the analyses of mating systems and genetic diversity [2,14,16]. Codominant loci usually provide a higher level of information per locus than dominant ones [17]. The SSR molecular marker is supposed to be the most suitable for studying mating systems and genetic diversity of populations [18,19,20]. Their successful application in resolving the genetic relationships of certain tree species has been extensively documented [4,21,22].

*Ormosia hosiei* (Leguminosae) is an endangered multipurpose subtropical tree species, commonly known as near rosewood, with typical papilionaceous, which is pollinated by insects and seeds spread by gravity, wind, and rats. It has been extensively used for ornamental purposes, bodywork and parquet. Due to its adaption to cold, *O. hosiei* is very useful for reforestation as a valued timber tree in central and northern subtropical regions of China. Recently, the number of natural populations of *O. hosiei* has declined rapidly, most of the extant natural populations are small with a size from several to dozens of plants as a consequence of overharvesting [23,24]. Field investigation found that even an isolated tree could blossom and fruit, which proposed that self-fertilization or apomixis may exist in the mating system. In addition, a certain amount of albino seedlings (about 4%) were found in the progeny [25], which confirmed the existence of inbreeding depression. Additional studies have investigated the genetic diversity by ISSR or SSR, showing a high level of genetic diversity of *O. hosiei* [23,24,25,26]. Thus, investigating outcrossing rates of *O. hosiei* in different environments, as well as heterogeneity of mating system, is of great importance for conservation, genetic improvement and artificial plantation establishment purposes of a tree species. However, to our knowledge, no study has been undertaken to estimate mating systems both in populations and single families of *O. hosiei*.

In this work, two populations and their progenies of *O. hosiei* were studied and the goals of this study were (1) to investigate and compare the genetic diversity in adult and progeny cohorts, (2) to estimate outcrossing rates in two natural populations using multi-locus mixed mating system model, (3) to test for any difference among individuals in two populations. This study is expected to provide a theoretical basis for the formulation of conservation strategies, breeding of improved varieties, and further development and utilization of *O. hosiei*.

## 2. Materials and Methods

### 2.1. The Species

The study species is *O. hosiei* (Fabaceae), listed as nearly threatened in the latest IUCN red list (http://www.iucnredlist.org/speciess/32432/9706557 (accessed on 1 January 1998)). *O. hosiei is* a tree with the highest economic value, widely distributed throughout southern of China, and pollinated by insects, dispersed of seeds by gravity or animal [27,28]. *O. hosiei* can grow up to 20–30 m and 100 cm of diameter at breast height (DBH), and is valued commercially for its timber, which is suitable for construction and furniture manufacturing, and its heartwood has also led to its use in sword. *O. hosiei* has bisexual flowers in panicles, stamen separation fully developed, or 1–5 of them without anthers. Flowers bloom mainly during the April to May months. The fruits are dehiscent pods 3.3–4.8 cm long and 2.3–3.5 cm wide, and contain one or two seeds (one kg of fruit contains about 900 seeds) [26]. Fruits usually ripen between October to November, and gravity and flowing water are the main seed dispersion vectors, *Niviventor confucianus* also have a certain effect on the dispersal of their seeds [28]. The adult plants of *O. hosiei* will not flower and bear fruit every year, and some will be separated by 3–5 years, or an even longer interval [29]. The size of natural population of *O. hosiei* varies from several to dozens, in which only a few plants or even one plant flower in a year; sometimes isolated trees can also flower.

### 2.2. Study Sites

According to the above phenomenon, two populations of *O. hosiei* were collected in South China, they were located in two geographically isolated small watersheds and in two different groups according to the results of structure analysis [29]. The first population, referred to here as JXMTS, was located in Zixi country (ZX), Jiangxi Province (northeast of Zixi country, 27.79° N/117.13° E; Figure 1). This region has an average altitude of 220 m, a wet subtropical climate, an average annual temperature of 16.9℃ and rainfall of 1930 mm per year. Most *O. hosiei* individuals in this area were remnants of forest fragmentation, while other immature individuals are naturally renewed. There are approximately 48 trees of *O. hosiei* in this area distribution along the Nangang river, which belong to the upper reaches of Luxi River, and some trees isolated at distances up to 1 km. The second population, called ZJBD, situated in Longquan city (LQ), Zhejiang Province (southwest of Longquan city, 29.03° N/119.99° E; Figure 1). It has a subtropical climate with an average annual temperature of 17.6℃ and annual precipitation of 1699 mm. In this area, there are 34 adult *O. hosiei* trees and some regeneration from root of the species was found. Most *O. hosiei* adult individuals are remnants of forest fragmentation, while others were planted.

### 2.3. Sample Collection

*O. hosiei* leaves were collected from 52 adult trees from two populations: 18 from JXMTS population and 34 from ZJBD population. The distances between the sampled trees were very different from each site, with greater distances for the ZJBD site than for the JXMTS population. The distance between sampled trees for ZJBD population ranged from 3 to 3183 m, with an average of 745 m. For the JXMTS population, the distances ranged from 42 to 2274 m, with an average of 633 m. In addition, 1 kg naturally pollinated seeds were collected form 4 and 3 maternal trees in ZJBD and JXZX population, respectively. In the population of JXMTS, its size was about 50 trees from which only those accessible to be collected were sampled, which included 3 maternal plants. In ZJBD population, only 4 maternal trees were gathered, while majority of others were scattered but had no fruits. In this case, all trees with fruits were sampled. In the following year, healthy leaf tissues of 34 adult trees (including 4 maternal trees) in ZJBD populations and 18 (including 3 maternal trees) in JXZX population were collected when the new leaves fully unfolded, and the color changed from red to green. Distribution of sampled adult trees was shown in Figure 1.

The seeds collected from maternal trees were sown in the nursery of Longquan Forestry Academy of Zhejiang Province, China (28.03 N/119.09 E). According to the hypothesis of Sjögren and Wyöni [30], at least 30 individual samples should be collected in order to detect rare alleles in a population (frequency < 5%). When the progeny seedlings reached approximately 15–20 cm height, leaf tissues were then collected from the seedlings for DNA isolation. Due to low germination in some families, 21–30 seedlings were genotyped per family (randomly selected 30 individuals or all progeny seedlings for less than 30 individuals). In this study, we referred to seedlings from a single maternal tree as progeny array or family. As in ZJBD population, totaling 120 progenies distributed in 4 families, while the total progenies were 77 in 3 families (Appendix A).

The leaf of progenies sampled were frozen in liquid nitrogen and stored at −80 °C until DNA extraction.

### 2.4. Genomic DNA Extraction and SSR Amplification

Leaves with removed petioles and veins were ground to a fine powder in liquid nitrogen and placed in microtubes. Total genomic DNA was isolated from 0.1 g of the fresh leaf tissues collected from both maternal trees and the progenies using the protocol described by Li et al. [29]. DNA was quantified on a Nanodrop 2000 spectrophotometer (Thermo Fisher Scientific, Massachusetts, USA).

Polymerase chain reaction (PCR) amplifications of 12 polymorphic nuclear microsatellite markers developed and optimized for *O. hosisei* by Li et al. [29] were used to genotype both the seedlings and the maternal trees. The DNA samples were amplified in the Takara PCR Thermal Cycler Dice Touch (TaKaRa, Dalian, China) with the procedure described by Li et al. [29].

PCR products were first detected on 2% agarose gel in 0.5× TBE buffer (100 V for 15–20 min) and then the capillary fragment electrophoresis of PCR products was scored against and automatically sized with the 20–1000 bp Alignment marker (Bioptic.inc) on a Qsep100TM automatic nucleic acid protein analysis system (BiOptic, Taiwan, China), using the Q-Expert software package (BiOptic). The size of SSR bands scored ranged from 50–400. All bands were set to a width of 2–6 bases pairs according to repeat motif, and the fragments with peak heights below 50 relative units were assumed to represent instrument noise and were not scored. To ensure that bands positions were assigned accurately, all bands were then checked manually. Any bands possessing fragments that overlapped with adjacent bands were removed. In addition, we adjusted bands assigned off-center of any peak distributions.

### 2.5. Data Analysis

#### 2.5.1. Estimation of Genetic Diversity Parameters

To check genotyping errors, we applied Micro-Checker [31] to discriminate errors arising from short allele dominance (large allele dropout), stuttering, and null alleles, according to the methods of Zhou et al. [32]. Using GenALEx6.5 software [33], the genetic diversity of populations (adults and progenies form both populations) and adult and progenies separately, were detected by means of the total number of alleles per locus (Na), effective number of alleles per locus (Ne), observed heterozygosity (Ho), and expected heterozygosity (He) under Hardy–Weinberg equilibrium. When the null allele existed in a locus, the expected heterozygosity (He) was calculated by $1-\sum kP_{ijk}^{2}$ where the null allele frequency was included and *P*_ijk_ was the frequency of allele k of locus j at population i. Estimates of allele frequencies were shown in Appendix A. Allele richness (A_R_) was obtained by HP-RARE 1.1 [34]. The fixation index (F), estimated to assess inbreeding in the samples, was calculated for maternal trees and the progenies in each population. The statistical significance of F values was tested using permutations (10,000 resampling) and the FSTAT program [35].

The population genetic differentiation was calculated by a program in R based on allele frequencies (Appendix A). For single locus estimate, we used the following formula:(1)$$F_{st(s)}=\frac{\sum_{i}\sum_{j}n_{i}{\left(P_{ijk}-{\bar{P}}_{jk}\;\right)}^{2}/\left(r-1\right)\bar{n}}{\sum_{j}{\bar{P}}_{jk}\left(1-{\bar{P}}_{jk}\;\right)}$$
where $\bar{P}$*_jk_* be the average frequency of allele *k* at locus *j* over *r* population, and $\bar{n}$ = $\sum_{i}n_{i}$/*r*. Chi-square statistic (χ2 = (*r* − 1) $\bar{n}F_{st(s)}$ with df = *r* − 1 was used to test the null hypothesis H0:Fst(s) = 0 [36]. For multi-locus estimate,
(2)$$F_{st(s)}=\frac{\sum_{i}\sum_{j}\sum_{k}n_{i}{\left(P_{ijk}-{\bar{P}}_{jk}\;\right)}^{2}/\left(r-1\right)\bar{n}}{\sum_{j}\sum_{k}{\bar{P}}_{jk}\left(1-{\bar{P}}_{jk}\;\right)}$$

The presence of null alleles was determined using Micro-checker version 2.2.3 software [31] and screened for genotyping errors before subjecting the data to statistical analysis. Hardy–Weinberg equilibrium (HWE) was tested using Genepop on the web (http://genepop.curtin.edu.au/ (accessed on 8 December 2020)) by a Markov chain dememorization of 1000, 100 batches, and 1000 iterations per batch [37,38]. The data were also tested for genotyping errors resulting from short allele dominance. The levels considered in the hierarchical analysis were populations, families within populations, and individuals as nested levels. Signification for variance components and F statistic were obtained by 1023 permutations. Additionally, genetic structure within each population was evaluated by means of analysis of variance components.

#### 2.5.2. Estimation of Mating System Parameters

The mating system of *O. hosisei* was estimated using known maternal parents. Estimates of multi-locus (t_m_) and single locus outcrossing rate (t_s_), fixation indices of maternal parents (F_M_), correlation of tm among progeny arrays (r_t_), and correlation of outcrossed paternity (r_p_) [including multi-locus paternity correlation (r_p(m)_) and single-locus paternity correlation (r_p(s)_)] were calculated using the mating system program (MLTR), described by Ritland [38], based principally on mixed mating model of Ritland and Jain [39]. Biparental inbreeding was also estimated following Ritland [40], as tm-ts. Each parameter was examined both between populations and among trees within a population. Analyses at two levels were carried out using the expectation maximization (EM) algorithm with the same default parameters as described by Pometti et al. [41] in the analysis of *Acacia visco*. The 95% confidence interval (CI) of each parameter was obtained from 1000 bootstrap replicates. The bootstrap analysis at the population level was carried out using adult trees as the units of resampling and at the family level using individuals with families as the resampling units.

In the above analyses, mating system parameters (t_m_, t_s_, r_p_) were used to estimate other demographic and genetic parameters (Nep). The effective number of pollen donors over maternal trees was calculated from the paternity correlation by the equation Nep = 1/rp (m) [39].

## 3. Results

### 3.1. Genetic Diversity and Mating System

#### 3.1.1. Genetic Diversity

All 12 microsatellite loci analyzed were polymorphic, with a total of 117 alleles observed in total sample (246). The number of alleles, heterozygosity and fixation index exhibited a rather high variation among locus (Table 1). In the fixation index (F) among the 12 SSR loci, only 5 (SSR1, SSR 7, SSR8, SSR 10, and SSR 13) were negative, while that of other loci were positive, indicating that most loci exhibited an obvious absence of heterozygote. 

Comparing adults and progenies from both populations, the average number of alleles was lower in adults than observed in progenies for both populations and a slightly higher in JXMTS than that in ZJBD population (Table 2). The observed heterozygosity was significantly higher in adults than progenies from both populations. The expected heterozygosity (with the null alleles counted) in two populations were greater than the observed heterozygosity and were similar in two generations and populations. The allelic richness (A_R_) in progeny-JXMTS was vaguely lower, whereas in terms of ZJBD population and adult-JXMTS, it was similar. The average fixation index (F) was positive and significantly different from zero in all populations, ranging from 0.217 (*p* <0.05) in adults to 0.484 (*p* <0.05) in progeny from JXMTS, showing inbreeding.

Analysis of population genetic differentiation with Equations (1) and (2) was summarized in Appendix A. Generally, population genetic differentiation was significant at each locus except SSR2, SSR5, SSR6, SSR8, and SSR13, ranging from F_st_ = 0.030 at locus SSR2 to 0.162 at SSR1. Genetic differentiation was also significant between populations and progenies except between adults-JXMTS and progenies-JXMTS (*p* = 0.051), and adults-JXMTS and progenies-ZJBD, (*p* = 0.059), the genetic differentiation between adults-JXMTS and adults-ZJBD was highest (Fst = 0.115), with the lowest (Fst = 0.041) between adults-JXMTS and progenies-ZJBD.

#### 3.1.2. Mating System

According to the instruction provided in MLTR, null alleles at relevant loci in both populations and each family were counted in estimating the outcrossing rate. Results are shown in Table 3 and Table 4, respectively. The estimate of multi-locus outcrossing rate (t_m_) and single-locus outcrossing rate (t_s_) was high in both populations (Table 3). These results indicated that both populations of *O. hosiei* are mostly outcrossers. The mean multi-locus rate (t_m_) was higher than the single-locus outcrossing rate (t_s_). The estimated biparental inbreeding rate (t_m_-t_s_) was low in two populations, and the value in the ZJBD population was greater than in the JXMTS population and significantly different from zero, suggesting some tendency to mating between relatives in two populations and a higher inbreeding rate existing in the ZJBD population than that in the JXMTS population.

The correlation of tm within progeny arrays (r_t_) differed between the two populations. For ZJBD population, the rt estimate was high (r_t_ = 0.910). However, for JXMTS population, the rt estimate was low (r_t_ = 0.108), it is compatible with the limited variations estimated tm among progenies (Table 4), indicating no differences in outcrossing rates among maternal plants. The correlation of outcrossed paternity (r_p(m)_ or r_p(s)_) was significant in JXMTS (r_p(m)_ = 0.142, Se = 0.057) and in ZJBD (r_p(m)_ = 0.213, Se = 0.067; r_p(s)_ = 0.276, Se = 0.087), suggesting a high probability that a randomly chosen pair of progenies from the different arrays was closely related. The obvious difference between r_p(m)_ and r_p(s)_ (r_p(s)_-r_p(m)_ = 0.063, Se = 0.026) in the ZJBD population indicated the existence of substructure. The effective number of pollen donors was 7.042 in the JXMTS population and 4.695 in the ZJBD population. Thus, the result may be interpreted, assuming that all plants blooming in the year of seed collection would act as pollen donors. In all cases, fixation index in progenies (F_o_) was positive, ranging from 0.072 to 0.094, indicating a certain extent of inbreeding in two populations. Fixation in adult trees (F_m_) was similar with that estimated for their progenies, but was significant from zero in two populations, suggesting Hardy–Weinberg deviations in the populations.

Generally, all the families showed high tm values in both populations with low variations among the families, ranging from 0.997(ZJBD3) to 1.000(JXMTS) (Table 3). The estimated biparental inbreeding rate (t_m_-t_s_) ranged from 0.000 (JXMTS1) to 0.139 (ZJBD2), with the mean of 0.047 over the seven families, suggesting some tendency to mating among families. The positive correlation of paternity at multiple loci was significant in ZJBD1, ZJBD2, and ZJBD3, indicating that the same fathers in pollination were partially shared in these families but not in JXMTS and ZJBD4. The correlation of outcrossed paternity (r_p(m)_) exhibited a quite high level of variation among families; family ZJBD2 had the highest value (0.352) and family JXMTS2 had the lowest value (0.097). The effective number of pollen donors (Nep) was greater than that estimated in populations, but had the same ranking patterns as Nep in populations.

## 4. Discussion

### 4.1. Genetic Diversity

Genetic diversity is the sum of genetic variations among all individuals in a species, which is critical to their long-term survival and reflects the adaptability of a species to the environment and its potential for modification [21]. In this study, we investigated the genetic diversity of adult and progeny generations of *O. hosiei*, a species endemic to southern China, by using nuclear microsatellite markers. Mountains, rivers and even forest fragments are often rich in endemism, but these habitats are commonly at risk of degradation, causing a reduction of genetic diversity of its populations [42,43]. Here, *O. hosiei* is endemic to the southern region of China, and most, if not all, populations of species were distributed along the river, they are in a highly particular environment with precipitation and anthropogenic influence, but we did not find low genetic diversity in the populations sampled adults and progeny cohorts (Table 1 and Table 2). We observed a substantial level of diversity (He = 0.763–0.794) that was higher than that of *Embothrium coccineum* (He = 0.21), a very common tree in relatively recently fragmented landscapes of southern Chile and a self-incompatible species, and equivalent to that of *Acacia pennata* (He = 0.57), *Copaifera langsdorffii* (He = 0.85), and *Dalbergia nigra* (He = 0.74), which belong to Leguminosae [16,44,45]. Although a direct comparison of genetic diversity levels between species is complicated by the life-history traits of different species, the levels of diversity calculated in *O. hosiei* can also be considered high, which is consistent with the studies that were based on ISSR [23,24]. These results suggest that the genetic diversity of *O. hosiei* populations, even though small with only a few adult plants, has not yet been influenced by habitat fragmentation and the level of genetic diversity is not necessarily synonymous with the rarity or endemicity of a species. Similarly, genetic diversity of *O. hosiei* did not vary with current population size, which may be related to longevity and recent fragmentation. Moreover, although the species’ clonality has not yet been formally studied, we observed evident clonal reproduction on our field investigation.

The genetic diversity of progenies (estimated from He, with null alleles counted) was higher than that of adults, but with no significant difference, which caused by the number or density of plants flowering in bearing year. Variation in density of plants flowering within populations may cause fluctuation of outcrossing rate and gene flow. The progeny cohorts produced by populations with clustered plants flowering may be lower outcrossing rate than cohorts produced by populations with high incidence of flowering. This is due to the expression of lethal recessive alleles when homozygosity increases in low density flowering plants or in small populations [46]. Many researchers observed that environment selection and inbreeding depression were beneficial to heterozygotes, leading to the excess of heterozygotes, such as *Berchemiella wilsonii* and *Tsoongiodendron odorum* [18,47]. In contrast to He, allelic richness (A_R_), a measure of genetic variation somehow different from gene diversity, is more sensitive to bottlenecks, demographic changes, reflects better part fluctuations in population size and is highly dependent on effective population size [48,49,50]. In the study of ours, both He and A_R_ were all relatively high, although, according to A_R_, the genetic diversity of the adults was slightly lower than that of the progenies. Our results showed an excess of homozygotes in both parent and progeny cohorts (F = 0.217~0.484), and the proportion of homozygotes among progeny cohorts was higher than that among parents. We observed a certain proportion of albino seedlings in progeny cohorts, which strongly suggests the existence of inbreeding depression. It should be noted that the progeny populations analyzed in this study were not grown in completely natural conditions, and the seeds were mechanically processed before germination and germinating in appropriate environmental conditions, which excluded the inbreeding depression during seed set and germination. Therefore, we underestimated the extent of inbreeding depression, which resulted in a higher proportion of homozygotes.

Previous studies demonstrated that population differentiation of *O. hosiei* was moderate, with F_st_ = 0.073 among watersheds with SSR and 0.116 among nine populations [29], and with F_st_ = 0.095~0.156 [23,24]. The present study also indicated small genetic differentiation (0.068) among adult and progeny populations. Low genetic differentiation between adults (F_st_ = 0.115), adults and progenies (F_st_ = 0.041~0.094), and progenies (Fst = 0.060) suggests that there is little restriction of gene flow (Nm > 1) among populations. Nevertheless, within-population inbreeding seems to follow distinct patterns in adults and progenies. The adult JXMTS population exhibited a higher fixation index (0.118) than progenies (0.072), while there was an opposite phenomenon in the ZJBD population, perhaps reflecting pre-fragmentation Wahlund effects and thus the existence of a former population structure. Reduced fixation index of progeny in the JXMTS population corelates with the observed high outcrossing rates.

### 4.2. Mating System

Information on mating systems is of fundamental importance for developing strategies for rational use and conservation programs for any species, because the distribution of genetic variation within and among populations and their inbreeding rates are highly dependent on their mating systems [41]. The outcrossing rates at the population level and among individuals within each population were similar between the two populations of *O. hosiei*. The multi-locus outcrossing rate obtained in the present paper are comparable to those reported for other leguminous species with different molecular markers as well as for different levels (Appendix A) [2,4,13,14,15,16,41,51,52,53,54]. The general trend observed is that Leguminous species, whether at the individual, population, species, or genus level, displayed a predominantly outcrossing system, with high t_m_ estimates. The high outcrossing rate depends on factors such as self-incompatibility, protogyny, and protandry, and selective abortion of self-pollinated fruits or seeds [2,14,51]. Studies have shown that self-incompatibility is considered an important outbreeding mechanism in Leguminous species [4,15,55]. However, whether *O. hosiei* is self-incompatible is still unknown. On the one hand, the few albino seedlings that appeared among the progenies may be a consequence of homozygous recessive deleterious alleles that typically arise from inbreeding, which indirectly proved that its mating system was not completely self-compatible. On the other hand, studies on other related species of *O. hosiei*, such as *E. fordii* and *R. pseudoacacia*, showed that flowers are dichogamous, with pollen grains coated with thick sticky substances and a stigma lacking a structure for capturing pollen and pollination by insects [15,55]. It was speculated that a high outcrossing rate observed in the present study related to the abovementioned mechanisms.

The comparison between multi-locus (t_m_) and average single-locus outcrossing rate (t_s_) can be used to infer inbreeding other than self-fertilization. Single-locus outcrossing rates are expected to be biased by any inbreeding in addition to self-fertilization. Therefore, the mean of such single-locus outcrossing rate is estimated to be lower than the multi-locus rate when mating among relatives occurs [56]. Comparison of tm-ts in *O. hosiei* indicated that this expectation was generally positive with the value of 0.000–0.139. The positive difference between t_m_ and t_s_ may be explained by mating among relatives or any other kind of genetic structure, which may be caused by the seed dispersal system. On the one hand, studies have shown that the seed of *O. hosiei* are spread by rats, water and gravity. Water flow and Edward’s long-tailed rats (*Leopoldamys edwardsi*) make a positive effect on seed dispersal, due to their accumulation and distribution of large quantities of seeds, while gravity and Chinese white-bellied rats (*N. confucianus*), hoard even more seeds in their nests, leads to a high relatedness of neighbor trees that are the most probably involved in reciprocal mating [26].

Considerable differences in outcrossing rates were detected among the investigated families. The multi-locus outcrossing rate (t_m_) and single-locus outcrossing rate (t_s_) of families in the JXMTS population was higher than that of in ZJBD population, which may be related to a few factors: different environmental factors or different pollinators; genetic relationship and geographic distance between mother plants and pollen donors [31], which predetermined the diversity of pollen [57], the distance between bearing plants is shorter in the ZJBD population than in that of the JXMTS population; and different proportions of adult plants and flowering plants in the two populations (six flowering plants in ZJBD population in 2015, which accounted for 17.65% of the adult plants, and five flowering plants in JXMTS population, accounting for 27.78%). It can be concluded that higher proportion of flowering is correlated with higher outcrossing rate in *O. hosiei*, but this is not always the case in other species. For example, species such as *P. elegans* mainly blossom annually, but some plants do not blossom year after year; thus, the proportion of flowering plants to adult plants ranged from 6% to 80%, and it was 33% and 80%, respectively, for two consecutive years in the same population, but the outcrossing rates were almost unchanged [58]. The differences in tm and ts among families of *O. hosiei* were similar to studies on *Magnolia stellata* and *T. odorum* [18,59], the differences were thought to be related to incomplete synchronization of flowering periods among individuals and different degrees of inbreeding decline in embryonic developmental stages, such as before seed maturity, germination of seed, and growth and development of individuals. Similarly, under natural conditions, the germination rate of *O. hosiei* seeds is only 3.4–5.1%, but after mechanical breaking, it can be as high as 94.7%. We also observed that the proportion of albino seedlings differed in different families, ranging from 0% to 13.2%, which indicated that an inbreeding decline occurred at different development stages. In Pinus pinaster, the size and crown of the seed trees were the main factors that affected the outcrossing rate among individuals. In this study, the breast diameter ranged from 37.3 cm to 125.5 cm and the crowns varied greatly, which could have resulted in the differences in the outcrossing rates among individuals. 

Significant differences in pollen and ovule allele frequencies were detected for all loci in all families of *O. hosiei* (Appendix A). These observed differences in allele frequencies between the ovule and pollen pool may be associated with the relative discrepancies in male and female function among trees, migration of pollen from outside of the population, selection between the time of pollination and progeny sampling, or most probably, non-random mating of genotypes outcrossing events [41,56]. In addition, fewer genotypes of mother plants may expect a bias in ovule allele frequencies, which may explain differences between ovule and pollen allele frequencies. The consequences caused by variation in allele frequencies are still incalculable, but as described by Ritland and Jain [40], it has minor effect on multi-locus estimated of outcrossing rate. Therefore, the estimation of outcrossing rate in this study was reasonable, despite relative difference allele frequencies between the ovule and pollen pool.

The effective number of pollen donors (Nep) per family suggests that few paternal parents contributed pollen to each progeny in the analyzed generation, indicating inbreeding or biparental inbreeding. The correlated mating found in the *O. hosiei* fragmented population may be associated with: (1) few plants flowering in the year of seed collection and low population density of the species, at three individuals per hectare (Investigated by author); few plants flowering and low population density is assumed to be related to reducing pollen diversification [5,54], leading to an increase mating between a limited number of individuals; (2) the limited number of maternal trees sampled, including trees that are very close to each other (Figure 1), resulting in increased mating between a limited number of individuals.

## 5. Conclusions

Despite fragmentation of *O. hosiei* in populations, both adults and progenies of *O. hosiei* had high genetic diversity, indicating that habitat fragmentation and anthropogenic disturbance could not have a significantly negative effect on these populations. Results from the multi-locus outcrossing rate of *O. hosiei* generally exhibited a predominantly outcrossing system. However, inbreeding may occur, as indicated by the comparison between tm and ts. The outcrossing rates varied among populations and individuals, which may be a consequence of different pollinator behavior caused by differences in environmental conditions and differences in pollen sources within the populations (density of adult trees). Limited seed dispersal and close genetic relationships of individuals in a smaller range lead to the occurrence of inbreeding, therefore, we suggested effective in situ conservation by artificial plantation establishment. For rehabilitation of disturbed populations, such as ZJBD and JXMTS, enrichment planting should be undertaken with materials collected from as many maternal trees as possible to prevent inbreeding depression due to possibility of mating among close relatives. The suggestion will not completely avoid inbreeding in the samples because inbreeding is determined by the natural mating system, which is out of our control. However, part of the inbreeding may be controlled at the stage of collecting seeds and seed nursery, by increasing the distance of seed collection and evicting albinism, low vitality, or other abnormal seedlings. Our results also imply that it is necessary to prevent human destruction of marginal or isolated individuals that will result in gene flow obstruction. Since we only analyzed seven families in two natural populations in a single year, it is necessary to conduct further multiple and multi-year experiments to verify our conclusions.

Therefore, for typical fragmented populations of other species, such as species with high outcrossing rate and genetic diversity, carrying out effective in situ protection by artificial plantation to promote natural regeneration and expansion of the existing population is proposed.

## Figures and Tables

**Figure 1 genes-13-02117-f001:**
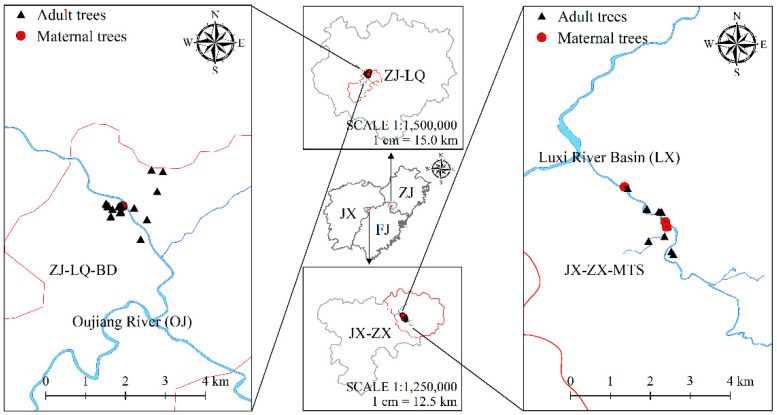
Geographic location of the sample collection areas of the *O. hosiei* adults (filled black triangle) and maternal trees (filled red circle).

**Table 1 genes-13-02117-t001:** Characteristics of the 12 microsatellite loci analyzed in samples of adults and progenies from two populations of *O. hosiei*.

	Adults-ZJBD (*n* = 34)					Adults-JXMTS (*n* = 15)				
	Na	Ne	Ho	He^1^	Fa	Na	Ne	Ho	He^1^	Fa
SSR1	4	3.229	0.941	0.691	−0.363	2	2.000	0.990	0.500	−0.385
SSR2	7	5.172	0.618	0.829	0.234	7	5.000	0.800	0.800	0.000
SSR3	11	7.049	0.529	0.868	0.383	9	4.737	0.333	0.824	0.577
SSR4	8	5.402	0.412	0.837	0.495	8	5.294	0.467	0.847	0.425
SSR5	8	4.489	0.206	0.784	0.735	7	4.327	0.533	0.827	0.306
SSR6	9	7.225	0.294	0.835	0.659	7	4.592	0.083	0.717	1.000
SSR7	8	5.389	0.882	0.814	−0.083	6	3.462	0.333	0.794	0.531
SSR8	8	6.067	0.667	0.862	0.202	5	3.383	0.653	0.704	−0.420
SSR9	4	2.431	0.265	0.690	0.550	7	5.114	0.733	0.804	0.088
SSR10	5	3.150	0.471	0.727	0.311	5	3.285	0.733	0.695	−0.054
SSR12	12	6.568	0.765	0.848	0.098	9	4.688	0.267	0.813	0.661
SSR13	4	3.336	0.647	0.700	0.076	7	5.921	0.933	0.831	−0.123
	**Progeny-ZJBD (n = 120)**					**Progeny-JXMTS (n = 77)**				
	**Na**	**Ne**	**Ho**	**He^1^**	**Fp**	**Na**	**Ne**	**Ho**	**He^1^**	**Fp**
SSR1	5	3.113	0.916	0.678	−0.349	5	3.762	0.909	0.734	−0.238
SSR2	7	4.787	0.517	0.822	0.347	8	5.174	0.636	0.847	0.211
SSR3	11	5.003	0.233	0.801	0.708	8	3.937	0.208	0.785	0.721
SSR4	8	5.666	0.592	0.846	0.282	8	3.751	0.189	0.769	0.742
SSR5	8	2.872	0.158	0.731	0.757	8	2.460	0.169	0.718	0.716
SSR6	9	5.426	0.034	0.737	0.959	6	3.901	0.013	0.715	0.982
SSR7	9	4.243	0.558	0.804	0.269	8	3.140	0.416	0.770	0.390
SSR8	10	6.566	0.975	0.848	−0.150	8	4.002	0.675	0.750	0.100
SSR9	6	4.241	0.442	0.808	0.422	10	3.152	0.403	0.771	0.410
SSR10	6	4.517	0.425	0.809	0.454	6	3.711	0.156	0.759	0.787
SSR12	10	7.573	0.142	0.786	0.837	10	5.313	0.091	0.758	0.888
SSR13	8	5.781	0.533	0.852	0.355	9	6.448	0.766	0.845	0.093

n, the total sample size; Na, the total number of alleles; Ne, the number of effective alleles; Ho, the observed heterozygosity; He^1^, the expected heterozygosity was calculated according to the allele frequencies in Appendix A where null allele frequencies were estimated if present at a locus, Fa, fixation index of adults; Fp, fixation index of progenies.

**Table 2 genes-13-02117-t002:** Genotyping errors and genetic diversity in adults and progenies of two populations of *O. hosiei*.

Populations	Stuttering Loci	Large Allele Dropout	Loci with Null Alleles	Na	Ho	He^1^	A_R_	F
Adults-ZJBD	No	No	SSR2, 3, 4, 5, 6, 9, 10	7.333 ± 2.674	0.558 ± 0.239	0.791 ± 0.069	5.452 ± 1.401	0.275 ± 0.318
Progeny-ZJBD	No	No	SSR2, 3, 4, 5, 6, 7, 9, 10, 12, 13	8.083 ± 1.832	0.460 ± 0.292	0.794 ± 0.053	5.427 ± 1.076	0.408 ± 0.385
Adults-JXMTS	No	No	SSR3, 4, 5, 6, 7, 12	6.583 ± 1.929	0.594 ± 0.322	0.763 ± 0.098	5.452 ± 1.238	0.217 ± 0.438
Progeny-JXMTS	No	No	SSR2, 3, 4, 5, 6, 7, 9, 10, 12	7.833 ± 1.528	0.386 ± 0.296	0.768 ± 0.042	5.059 ± 0.772	0.484 ± 0.380

Na, the total number of alleles; Ho, the observed heterozygosity; He^1^, the expected heterozygosity was calculated according to the allele frequencies in Appendix A where null allele frequencies were estimated if present at a locus; A_R_, allelic richness; F, the fixation index.

**Table 3 genes-13-02117-t003:** Mating system parameters of two contrasting populations of *O. hosiei*.

Parameters	JXMTS	ZJBD
Adult trees (progenies)	15 (77)	34 (120)
Multilocus outcrossing rate (t_m_)	1.000 (0.055)	1.000 (0.285)
Single-locus outcrossing rate (t_s_)	0.969 (0.133)	0.869 (0.396)
Biparental inbreeding rate(t_m_-t_s_)	0.031 (0.133)	0.131 (0.051)
Correlation of outcrossing rate among progeny arrays (r_t_)	0.108 (0.000)	0.910 (0.118)
Multilocus paternity correlation (r_p(m)_)	0.142 (0.057)	0.213 (0.067)
Singlelocus paternity correlation (r_p(s)_)	0.404 (0.275)	0.276 (0.087)
r_p(s)_-r_p(m)_	0.262 (0.173)	0.063 (0.026)
Fixation index in adult trees (F_m_)	0.118 (0.086)	0.079 (0.133)
Fixation index in progenies (F_o_)	0.072(0.078)	0.094(0.065)
Effective number of pollen donors (N_ep_ = 1/r_p_)	7.042	4.695

*p* values are shown in parenthesis.

**Table 4 genes-13-02117-t004:** Mating system indices for the two contrasting populations of *O. hosiei* (JXMTS and ZJBD).

Maternal Tree	*n*	t_m_	t_m_-t_s_	r_p(m)_	N_ep_
JXMTS 1	21	1.000 (0.000)	0.000 (0.000)	0.110 (0.000)	9.09
JXMTS 2	26	1.000 (0.000)	0.032 (0.011)	0.097 (0.042)	10.31
JXMTS 3	30	1.000 (0.001)	0.020 (0.001)	0.101 (0.000)	9.9
ZJBD 1	30	1.000 (0.000)	0.041 (0.012)	0.215 (0.062)	4.65
ZJBD 2	30	1.000 (0.000)	0.139 (0.043)	0.352 (0.127)	2.84
ZJBD 3	30	0.997 (0.003)	0.055 (0.013)	0.240 (0.012)	4.16
ZJBD 4	30	0.999 (0.000)	0.040 (0.002)	0.099 (0.000)	10.1

*n*, family sample size; tm, multi-locus outcrossing rate; t_s_, single-locus outcrossing rate; t_m_-t_s_, biparental inbreeding; r_p(m)_, multi-locus paternity correlation; N_ep_, effective number of pollen donors; *p* values are shown in parenthesis.

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
