# Peer review of "Mating Systems of Single Families and Population Genetic Diversity of Endangered *Ormosia hosiei* in South China"

_genes, 2022, doi:10.3390/genes13112117_

Round 1
Reviewer 1 Report
Title. The title conveys the main message of the paper — the issues addressed and the relationships among the issues.
Abstract. The abstract is concise, provides a clear overview, includes essential facts for the paper, and concludes with a final point that places the work described in a broader context.
Keywords. These are enough for the topic.
Introduction. The introduction includes background to provide an appreciation for the context of the work presented and also states the rationale and problem that the researchers attempted to answer through their experiments.
Material and methods. In this section, the authors describe the correct steps that followed during conducting their study, give precise details of the study design, and how analyzed the data.
Results and discussion. This section was well written and shows all data with good descriptions. The results say about the objective that motivates the research, and the authors take a broad look at their findings and examine the work in the larger context of the field.
Conclusion. This section included the major conclusions, which were briefly written.
Figures.
Figure 1 — Authors must improve the quality because the letters are too small.
Tables.
Tables 1, and 7 — These tables must be moved to the supplementary materials.
Author Response
Major points
Figure 1— Authors must improve the quality because the letters are too small.
Tables 1, and 7 — These tables must be moved to the supplementary materials.
Response: Thank you for the comment.
- Related to the Figure 1.
We have improved the quality, and changed the letters from 12 to 14 and 16. If the letter is still small, we can adjust it again.
- Relate to the Table 1 and 7.
We have moved table 1 and 7 to supplementary materials in modified version.

Reviewer 2 Report
Authors are presenting a study of a rare and endangered tree species Ormosia hosiei in South of China. Study is focused on mating system analysis genetic variation of two populations (246 samples) from geographically isolated areas in South of China based on 12 polymorphic microsatellite markers. The main objectives of the study were to investigate and compare the genetic diversity in adult and progeny cohorts, to estimate outcrossing rates in two natural populations using multi-locus mixed mating system model, to test for any difference among individuals in two populations. Authors was expecting from the study to get a theoretical basis for the formulation of conservation strategies, breeding of improved varieties, and further development and utilization of O. hosiei.
Presented results showed that studied Ormosia hosiei populations have high level of genetic diversity in both adults and progenies, although a slightly differences existed among populations and their progenies, with the expected heterozygosity ranged from 0.763-0.794. Overall, O. hosiei displayed a predominantly outcrossed mating. The estimate of multi-locus outcrossing rate (tm) was high with low variations among families, ranged from 0.997 to 1.000. The value of tm – ts indicated that biparental inbreeding occurred in progenies. Based on the results authors have suggested that for effective in-situ conservation need to apply replanting, but seedling production for restoration purposes may require a much larger sampling effort than is currently used. And that it is necessary to conduct further multiple populations and multi-year experiments to verify our conclusions.
I would recommend for authors to apply allelic richness measure (the number of different alleles segregating in the population) as more informative and of key importance in conservation programmes, especially for subdivided and scattered tree species as an alternative criterion for measuring genetic diversity (Marshall and Brown 1975, Petit et al. 1998, Rajora et al., 2000, Rajora and Mosseler 2001, Caballero et al. 2010, etc.). In contrast to He, allelic richness is more sensitive to bottlenecks, reflects better past fluctuations in population size and is highly dependent on effective population size (Petit et al. 1998, Caballero et al. 2010 and references therein). Uneven distribution of allelic diversity (allelic richness) among and within populations can be applied for prioritisation of populations for conservation (Petit et al. 1998, Caballero et al. 2010).
Overall, the manuscripts tackle an interesting topic dealing with mating system analysis and conservation of endangered tree species. This topic has special focus in forest geneticists’ community; therefore, it is relevant and important. Sampling design is appropriate for mating system analysis, however from population genetics and conservation point of view sampling should be more extensive. Manuscript can be published by journal Genes after substantial improvements and corrections. I don't feel qualified to judge about the English language and style, however I think English proof reading for manuscript is necessary.
Literature survey also could be improved by literature sources referring to conservation genetics of rare/endangered tree species and referring additionally to important diversity parameter for conservation genetics such as allelic richness. There are number of studies which underlined the importance of Ar for conservation genetics. Suggested improvements for list of references:
Hughes RA, Inouye BD, Johnson MTJ, Underwood N, Vellend M (2008) Ecological consequences of genetic diversity. Ecol Lett 11:609–623.
Marshall, D. R. and Brown A. H. D. (1975). Optimum sampling strategies in genetic conservation. In: Frankel, O. H., and Hawkes, J. G. (Eds.). (1975). Crop genetic resources for today and tomorrow (Vol. 2). Pp.53-80. CUP Archive.
Petit, R.J.; Elmousadik, A.; Pons, O. (1998): Identifying populations for conservation on the basis of genetic markers. Conservation Biology, 12(4), S. 844 - 855.
Rajora, O. P., Mosseler, A. (2001). Challenges and opportunities for conservation of forest genetic resources. Euphytica, 118(2), 197-212.
Rajora, O. P., Rahman, M. H., Buchert, G. P., & Dancik, B. P. (2000). Microsatellite DNA analysis of genetic effects of harvesting in old‐growth eastern white pine (Pinus strobus) in Ontario, Canada. Molecular ecology, 9(3), 339-348.
Caballero, A., Rodríguez-Ramilo, S. T., Avila, V., & Fernández, J. (2010). Management of genetic diversity of subdivided populations in conservation programmes. Conservation Genetics, 11(2), 409-419.
Leberg, P. L. (2002). Estimating allelic richness: effects of sample size and bottlenecks. Molecular ecology, 11(11), 2445-2449.
- Other comments and suggested corrections are provided in the manuscript attached.

Author Response
Reviewer #2
- I would recommend for authors to apply allelic richness measure (the number of different alleles segregating in the population) as more informative and of key importance in conservation programmes, especially for subdivided and scattered tree species as an alternative criterion for measuring genetic diversity (Marshall and Brown 1975, Petit et al. 1998, Rajora et al., 2000, Rajora and Mosseler 2001, Caballero et al. 2010, etc.). In contrast to He, allelic richness is more sensitive to bottlenecks, reflects better past fluctuations in population size and is highly dependent on effective population size (Petit et al. 1998, Caballero et al. 2010 and references therein). Uneven distribution of allelic diversity (allelic richness) among and within populations can be applied for prioritisation of populations for conservation (Petit et al. 1998, Caballero et al. 2010).
Response: Thank you for the comment.
In the section of date analysis, we added the analysis method of allelic richness. And in the results and discussion, we also analyzed the differences of allelic richness in adult and progeny populations. Result displayed that both He and AR were relatively high, although, according AR, the genetic diversity of adults was slightly higher than that of progenies, whereas, the results were opposite in the light of He. Correspondingly, references have also been added. See the revised version for details.
Minor points
- In abstract section, the second sentence is not clear. And in-situ must be in italic.
Response: Thank you for the comment.
The second sentence: The mating patterns have substantial impact on genetic diversity and structure. It can be modified as: the mating patterns have substantial impact on genetic diversity and structure of plant natural population.
In situ has been modified to italic.
- In introduction section, “Therefore, knowledge of mating system and factors affecting mating system are essential in understanding the genetic structure of plant populations in anthropogenic disturbance that require restoration or conservation [9], due to its precondition for genetic improvement population conservation, and rational exploitation and utilization of resources.” It is a very long and complicated sentence. Need to rewrite.
Response: Thank you for the comment.
We modified it as follows: Therefore, the understanding of the mating system and factors affecting it is fundamental importance for genetic structure of plant populations in anthropogenic disturbance that require restoration or conservation [9]. Such information is useful in precondition for genetic improvement, population conservation, and rational exploitation and utilization of resources.
- “with the rapid development of molecular biology, molecular markers, such as isozyme markers, SSR, AFLP, and RAPD, have become widely used in the analyses of mating systems and genetic diversity. ” References needed.
Response: Thank you for the comment. We added references as follows: [2,14,16,53].
- Last sentences are complicated. Need to rewrite. And last sentence seems not finished ? (lines 77 to 83)
Response: Thank you for the comment.
The sentence was modified as follows: Thus, investigating outcrossing rates of O. hosiei in different environment, as well as relative importance of different population and heterogeneity of mating system, is of great importance for conservation, genetic improvement and artificial plantation establishment purposes of a tree species. However, to our knowledge, no study has been undertaken to exam genetic diversity and mating system parameters at individual and population levels.
- To long sentence, must be divided. And some repeating information. (lines 93 to 96)
Response: Thank you for the comment.
The sentence was modified as follows: The study species is O. hosiei (Fabaceae), listed as nearly threatened in the latest IUCN red list (http://www.iucnredlist.org/apps/redlist/details/32432/0). O. hosiei is a tree with the highest economic value, widely distributed throughout southern of China, and pollinated by insects, dispersed of seeds by gravity or animals.
- Unclear sentence. Need to rewrite. (lines 105 to 107)
Response: Thank you for the comment.
The sentence was modified as follows: The adult plants of O. hosiei don’t bloom and bear fruit every year, but every 3-5 year, even longer in some individuals.
- what is Ne, He1 and Fp? (the note of table 2 in the original manuscript)
Response: Thank you for the comment.
We added the note of Ne and Fp, such as Ne, the number of effective alleles; Fa, fixation index of adults; Fp, fixation index of progenies. The note of He is that of He1.
- Table 4 in original manuscript is to redundant, can go to annexes.
Response: Thank you for the comment.
We have moved table 4 to supplementary materials in modified version.
- Complicated sentence (lines 330 to 334). Must to rewrite.
Response: Thank you for the comment.
The sentence was modified as follows: Here, O. hosiei is endemic to the southern region of China, and most, if not all, populations of species were distributed along the river, which is a highly particular environment with precipitation and anthropogenic influence, but we did not find low levels of genetic diversity in the populations sampled across its geographical range and progeny cohorts.

Reviewer 3 Report
This study presents genetic diversity of Ormosia hosiei by molecular marker techniques currently used, thus please highlight the contribution of your study to the research progress in this field.
Author Response
Reviewer #3
This study presents genetic diversity of Ormosia hosiei by molecular marker techniques currently used, thus please highlight the contribution of your study to the research progress in this field.
Response: Thank you for the comment.
In conclusion section, we added as follows: Therefore, for typical fragmented populations of other species, it is proposed to carry out effective in-situ protection, artificially plantation to promote natural regeneration and expansion of the existing population.

Round 2
Reviewer 2 Report
Dear Authors,
Thank you for the authors for all the improvements of the manuscript. However, manuscript still should be improved. Find all the comments in the manuscript. In addition I strongly recommend for authors to perform English proof reading by native speaker. Because article soundness is quite low and sometimes it is hard to understand what authors wants to say. Sentences often to long and complicated.
Best regards.

Author Response
- In abstract section (Line 12-14), the second sentence is not clear. And in-situ must be in italic.
Response: Thank you for the comment.
The second sentence: The mating system has substantial impact on genetic diversity and structure of plant natural population. Such information should be considered when planning tree planting for forest res-toration
In situ has been modified to italic.
- In introduction section (Line 38-42). It is a very long and complicated sentence. Need to rewrite.
Response: Thank you for the comment.
We modified it as follows: Therefore, to the populations in anthropogenic disturbance and requiring restoration or conservation, the understanding of the mating system and factors affecting it is fundamental importance for genetic structure [9]. Such information is useful in precondition for genetic improvement, population conservation, and rational exploitation and utilization of resources.
- In line 61, References needed.
Response: Thank you for the comment. We added references as follows: [2,14,16,53].
- In lines 77 to 83, last sentences are complicated. Need to rewrite. And last sentence seems not finished?
Response: Thank you for the comment.
The sentence was modified as follows: Thus, investigating outcrossing rates of O. hosiei in different environment, as well as het-erogeneity of mating system, is of great importance for conservation, genetic improvement and artificial plantation establishment purposes of a tree species. However, to our knowledge, no study has been undertaken to estimate mating systems at single families and population genetic diversity of O. hosiei.
- In lines 93 to 96, it is a too long sentence, must be divided. And some repeating information. In addition, Reference for IUCN is missing.
Response: Thank you for the comment.
The sentence was modified as follows: The study species is O. hosiei (Fabaceae), listed as nearly threatened in the latest IUCN red list (http://www.iucnredlist.org/apps/redlist/details/32432/0). O. hosiei is a tree with the highest economic value, widely distributed throughout southern of China, and pollinated by insects, dispersed of seeds by gravity or animal [25,26].
Similarly, reference for IUCN was added.
- Unclear sentence. Need to rewrite. (lines 105 to 107)
Response: Thank you for the comment.
The sentence was modified as follows: The adult plants of O. hosiei will not flower and bear fruit every year, and some will be separated 3-5 years, even longer interval [28].
- In line 144, The last half sentence should form a new sentence.
Response: Thank you for the comment.
The sentence was modified as follows: Distribution of sampled adult trees was shown in Fig. 1.
- In line 189, Micro-Checker references?
Response: Thank you for the comment.
Reference was added, see modified version.
- In line 250-252, what is Ne, He1 and Fp? (the note of table 2 in the original manuscript)
Response: Thank you for the comment.
We added the note of Ne and Fp, such as Ne, the number of effective alleles; Fa, fixation index of adults; Fp, fixation index of progenies. The note of He is that of He1.
- Table 4 in original manuscript is to redundant, can go to annexes.
Response: Thank you for the comment.
We have moved table 4 to supplementary materials in modified version.
- Lines 330 to 334, it is a Complicated sentence. Must to rewrite.
Response: Thank you for the comment.
The sentence was modified as follows: Here, O. hosiei is endemic to the southern region of China, and most, if not all, populations of species were distributed along the river, they are in a highly particular environment with precipitation and anthropogenic influence, but we did not find low genetic diversity in the populations sampled adults and progeny cohorts (Table 1, 2).
- Line 351-354, so flowering or blooming? Same terms must be used over whole manuscript. Check introduction.
Response: Thank you for the comment.
We have checked the whole manuscript, and same terms were used.
See modified version for details.
